# Spatial Symmetry in Slot Attention

**Ondrej Biza***                                                    BIZA.O@NORTHEASTERN.EDU
*Northeastern University, Boston, MA, USA*

**Sjoerd van Steenkiste**                                          SVANSTEENKISTE@GOOGLE.COM
**Mehdi S. M. Sajjadi**                                               MSAJJADI@GOOGLE.COM
**Gamaleldin F. Elsayed**                                          GAMALELDIN@GOOGLE.COM
**Aravindh Mahendran**[†]                                          ARAVINDHM@GOOGLE.COM
**Thomas Kipf**[†]                                                     TKIPF@GOOGLE.COM
*Google Research*

**Editors:** Sophia Sanborn, Christian Shewmake, Simone Azeglio, Arianna Di Bernardo, Nina Miolane

## Abstract

Automatically discovering composable abstractions from raw perceptual data is a long-standing challenge in machine learning. Slot-based neural networks have recently shown promise at discovering and representing objects in visual scenes in a self-supervised fashion. While they make use of permutation symmetry of objects to drive learning of abstractions, they largely ignore other spatial symmetries present in the visual world. In this work, we introduce a simple, yet effective, method for incorporating spatial symmetries in attentional slot-based methods. We incorporate equivariance to translation and scale into the attention and generation mechanism of Slot Attention solely via translating and scaling positional encodings. Both changes result in little computational overhead, are easy to implement, and can result in large gains in data efficiency and scene decomposition performance.

**Keywords:** Spatial symmetry, Equivariance, Abstraction, Object-centric learning, Unsupervised learning

## 1. Introduction

Slot-based neural networks learn to represent inputs using a discrete number of latent vectors, often referred to as "slots". These are a promising class of architectures for learning object representations (Greff et al., 2020). In Slot Attention (Locatello et al., 2020), slots learn to describe the individual objects in an image through an iterative clustering procedure that leverages the permutation equivariance of objects. However, other inductive biases, such as equivariance to the location and scale of objects is absent, and thus must be learned in a potentially sample- and parameter-inefficient manner from input data alone. This is different from humans, who are believed to attach reference frames to objects to facilitate translation symmetric reasoning about objects and their parts (Hinton, 1981; Hawkins et al., 2019).

Spatial symmetries were successfully incorporated as an inductive bias to improve sample efficiency, generalization and consistency of predictions of neural networks (Thomas et al., 2018; Wang et al., 2020; Han et al., 2022). These advances have, however, had only a limited impact in object discovery. Prior object discovery methods often use monolithic

---

* Work performed while at Google Research.
† Equal contribution.

encoders (Eslami et al., 2016; Kosiorek et al., 2018) to process images and populate latent slots. Limited equivariance to translation is present in the case of convolutional encoders with output anchors (Crawford and Pineau, 2019; Lin et al., 2020). Some works employ spatial symmetries in the decoder (Eslami et al., 2016; Lin et al., 2020) using the Spatial Transformer (Jaderberg et al., 2015), which is equivariant to $2D-$affine transformations. Yet another popular choice, the Spatial Broadcast Decoder (Watters et al., 2019), breaks symmetry by appending absolute positions to pixels.

In this work, we explore the symmetry of object translation and object scale in Slot Attention (Locatello et al., 2020) applied to object discovery. We equip each slot with an explicit representation of position and a scale, and ensure that the same model weights can be used to detect and reconstruct objects at different positions and scales. Equivariance is achieved by encoding pixel positions relative to each slot both in Slot Attention and in the Spatial Broadcast Decoder. Although our model is not end-to-end equivariant (Cohen and Welling, 2016, 2017), as we use a standard convolutional encoder to allow for some flexibility in encoding absolute positions and scales of objects (Park et al., 2022), we find that it has better sample efficiency and generalization properties. Additionally, we find that the equivariant model is more likely to converge to favorable solutions, instead of collapsing to failure modes, such as always predicting Voronoi tessellated segmentation masks.

## 2. Equivariant Slot Attention

The key observation is that many slot-based models (Locatello et al., 2020; Singh et al., 2021; Sajjadi et al., 2022) and other scene representation approaches (Mildenhall et al., 2020; Sajjadi et al., 2021) append absolute ($2D$ or $3D$) positions to latent representations in order to encode and reconstruct images. These models are sensitive to positions and have to re-learn spatial symmetries from data. By giving slots explicit positions and scales, we can make position encodings relative to slots, making the model symmetric. Specifically, we propose translation and scale equivariant Slot Attention and Spatial Broadcast Decoder, but the same technique could be used with other models and symmetries.

**Slot Attention (SA)** (Locatello et al., 2020) computes cross attention between input tokens ($\texttt{inputs} \in \mathbb{R}^{N \times D_{inputs}}$) and latent slots ($\texttt{slots} \in \mathbb{R}^{K \times D_{slots}}$). The input tokens have an absolute coordinate grid, $\texttt{abs\_grid} \in \mathbb{R}^{N \times 2}$, attached to them. This makes the cross attention sensitive to positions. Keys and values for this are computed as follows using learned linear projections / MLPs $f, k, g, v$[1]:

$$\texttt{keys} = f(k(\texttt{inputs}) + g(\texttt{abs\_grid})), \quad \texttt{values} = f(v(\texttt{inputs}) + g(\texttt{abs\_grid})). \quad (1)$$

In contrast, in equivariant Slot Attention (Algorithm 1, see Appendix), we equip the $K$ slots with randomly sampled positions, $S_p \in \mathbb{R}^{K \times 2}$, and scales, $S_s \in \mathbb{R}^{K \times 2}$ and use these to create a separate *relative* coordinate grid for each slot:

$$\forall k \in \{1, \ldots, K\} \quad \texttt{rel\_grid}^k = (\texttt{abs\_grid} - S_p^k) \, / \, S_s^k \,. \quad (2)$$

---

1. Note that we add position embeddings after the key projection, this trick does not hurt SA's performance but makes equivariant slot attention more computationally efficient.

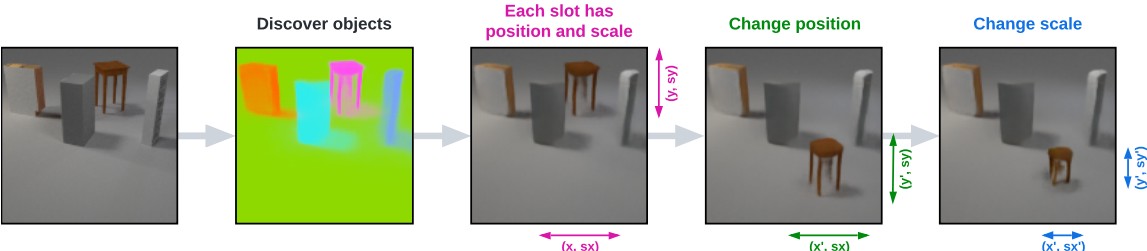

Figure 1: By combining unsupervised object discovery and explicit slot positions and scales, we can control how individual objects are decoded without any supervision.

By attaching this relative grid to the input tokens we effectively center and scale the input tokens into each slot's own coordinate frame. This achieves the desired spatial symmetry. In more detail, we replace (1) with the following:

$$\forall k \in \{1, \ldots, K\} \qquad \mathtt{keys}^k = f\left(k(\mathtt{inputs}) + g(\mathtt{rel\_grid}^k)\right)$$

$$\mathtt{values}^k = f\left(v(\mathtt{inputs}) + g(\mathtt{rel\_grid}^k)\right).$$

After the cross attention step, slots are updated as in standard SA. $S_p$ and $S_s$ are replaced by the center of mass and spread of the attention mask ($\mathtt{attn} \in \mathbb{R}^{N \times K}$), respectively.

$$S_p = \frac{\sum_n \mathtt{attn}_n * \mathtt{abs\_grid}_n}{\sum_n \mathtt{attn}_n}, \qquad S_s = \sqrt{\frac{\sum_n (\mathtt{attn}_n + \epsilon) * (\mathtt{abs\_grid}_n - S_p)^2}{\sum_n (\mathtt{attn}_n + \epsilon)}}$$

Similarly, we make the **Spatial Broadcast Decoder** (Watters et al., 2019) translation and scale equivariant: we compute the final slot positions and scales using the attention map of the last iteration of SA, create relative coordinate grids as in (2), and then add them to the broadcasted slots after applying a learned linear transformation. This ensures that an object can be decoded using the same weights at arbitrary sizes and scales (Figure 1).

## 3. Experiments

We evaluate equivariant Slot Attention across four synthetic datasets: Tetrominoes (Greff et al., 2019), CLEVRTex (Karazija et al., 2021), ObjectsRoom (Eslami et al., 2018) (in the Appendix), and CLEVR (Johnson et al., 2017) (in the Appendix). These datasets cover simple backgrounds with simple objects (Tetrominoes, CLEVR, ObjectsRoom) as well as fully-textured backgrounds/objects (CLEVRTex). We test (1) whether equivariant Slot Attention generalizes out of distribution if the data is fully symmetric to translations, and (2) whether incorporating spatial symmetries leads to overall better scene decomposition on standard multi-object benchmark tasks.

**Generalization and sample efficiency in Tetrominoes: A proof of concept** The Tetris-like objects in the Tetrominoes dataset have the same appearance regardless of their position (no occlusion, lighting or perspective changes); hence, Slot Attention should benefit from the inductive bias of translation equivariance. It achieves above 90% FG-ARI using

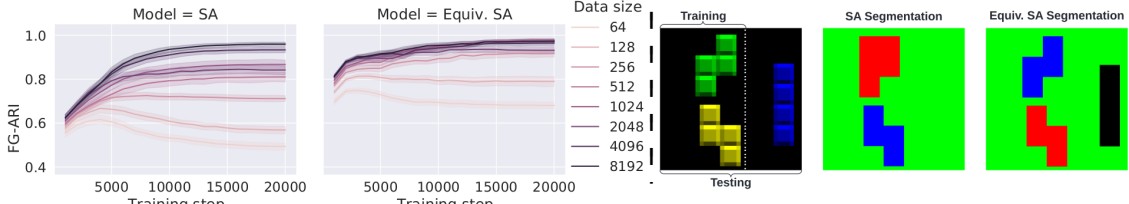

Figure 2: Tetrominoes dataset. Left: Translation equiv. Slot Attention achieves higher FG-ARI with less data. Right: T-SA generalizes better to OOD test-time configurations.

| Method | CLEVRTex | | CLEVRTex CAMO | | CLEVRTex OOD | |
|---|---|---|---|---|---|---|
| | ↑FG-ARI | ↓MSE | ↑FG-ARI | ↓MSE | ↑FG-ARI | ↓MSE |
| SPACE | $17.5_{\pm 4.1}$ | $298_{\pm 80}$ | $10.6_{\pm 2.1}$ | $251_{\pm 61}$ | $12.7_{\pm 3.4}$ | $387_{\pm 66}$ |
| DTI | $79.9_{\pm 1.4}$ | $438_{\pm 22}$ | $72.9_{\pm 1.9}$ | $377_{\pm 17}$ | $73.7_{\pm 1.0}$ | $590_{\pm 4}$ |
| Gen-V2 | $31.2_{\pm 12.4}$ | $315_{\pm 106}$ | $29.6_{\pm 12.8}$ | $278_{\pm 75}$ | $29.0_{\pm 11.2}$ | $539_{\pm 147}$ |
| eMORL | $45.0_{\pm 7.8}$ | $318_{\pm 43}$ | $42.3_{\pm 7.2}$ | $269_{\pm 31}$ | $43.1_{\pm 9.3}$ | $471_{\pm 51}$ |
| SimpleCNN SA | $54.5_{\pm 1.6}$ | $241_{\pm 14}$ | $53.0_{\pm 1.6}$ | $217_{\pm 12}$ | $54.2_{\pm 2.6}$ | $282_{\pm 12}$ |
| SimpleCNN T-SA | $66.8_{\pm 5.7}$ | $230_{\pm 20}$ | $65.0_{\pm 4.9}$ | $213_{\pm 16}$ | $65.1_{\pm 4.8}$ | $459_{\pm 25}$ |
| SimpleCNN TS-SA | $74.1_{\pm 6.4}$ | $224_{\pm 4}$ | $69.0_{\pm 5.4}$ | $210_{\pm 5}$ | $69.6_{\pm 4.3}$ | $471_{\pm 30}$ |
| ResNet SA | $80.8_{\pm 12.3}$ | $230_{\pm 45}$ | $74.3_{\pm 13.1}$ | $249_{\pm 34}$ | $74.3_{\pm 8.8}$ | $606_{\pm 45}$ |
| ResNet T-SA | $87.6_{\pm 4.0}$ | $198_{\pm 21}$ | $80.7_{\pm 3.9}$ | $223_{\pm 29}$ | $78.6_{\pm 3.3}$ | $611_{\pm 26}$ |
| ResNet TS-SA | $86.4_{\pm 9.4}$ | $219_{\pm 63}$ | $79.4_{\pm 9.9}$ | $244_{\pm 52}$ | $78.7_{\pm 7.0}$ | $625_{\pm 52}$ |

Table 1: CLEVRTex results on the test set, CAMO set (objects and backgrounds blend together) and OOD set (novel textures). Prior results taken from (Karazija et al., 2021) use 3 random seeds, we use 10 random seeds. FG-ARI is reported in %.

only one eighth of the dataset size required by the baseline SA model (256 vs. 4096). In Figure 2 (left), we perform a generalization experiment wherein we filter the training set for images with objects only appearing on the left side. The validation set is unchanged. We find that non-equivariant Slot Attention is less likely to detect objects on the right side of the image (FG-ARI $80.6 \pm 6.8\%$ compared to $94.8 \pm 1.5\%$ for T-SA), likely because it does not have any in-built inductive biases to promote generalization to unseen spatial configurations in the input.

**Comparison to state of the art on CLEVRTex** CLEVRTex is a challenging dataset with textured foreground objects and backgrounds. Previously, it was understood that Slot Attention cannot handle textures, as the FG-ARI score of 62.4% for the original Slot Attention (Table 1) is close to a naive Voronoi tessellation baseline (around 52% FG-ARI).

Our main finding is that the results of Slot Attention (SA) can be significantly improved by adding translation equivariance (T-SA). A further benefit can be observed by adding both translation and scale equivariance (TS-SA). This version of Slot Attention is trained using a simple 4-layer CNN backbone as in Locatello et al. (2020). We further find that replacing the simple CNN encoder of Slot Attention with a ResNet-34 (He et al., 2016)

backbone significantly improves scene decomposition performance (Table 1, SA (ours)). Adding translation equivariance further improves Slot Attention's ability to segment objects. Here, adding scale equivariance does not lead to significant further improvement, it does however enable explicit control of slot scales when decoding the learned slot representations. ResNet T-SA and TS-SA outperforms all baselines reported in Karazija et al. (2021) without pre-training. Sauvalle and de La Fortelle (2022) reported around 95% FG-ARI with a pre-trained SegFormer backbone (Xie et al., 2021) and a background model, which could be further combined with our approach.

## 4. Conclusion

We have introduced translation- and scale-equivariant Slot Attention. Our method enables incorporation of spatial symmetries with little computational overhead via simple changes to the positional encoding used both in the attention mechanism and the decoder of Slot Attention. We are excited about the potential of incorporating additional symmetries through similar mechanisms to a broader class of slot-based neural architectures.

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

## Appendix A. Limitations

- Our method struggles with small spatial input grids. We guess that this is because a sparsely sampled position grid results in a poor learning signal and unstable gradients.

- Computing keys and values per slot scales linearly with the number of slots. For large scale applications with 100s of slots this could cause memory issues.

- The real world is 3D and we only model 2D translation and scale symmetries in this work. Thus the input domain does not respect spatial symmetries 100%. We intentionally allow for global positions to leak through the CNN encoder so that the model may leverage these when necessary. We interpret this as a feature not a limitation of our model. A better solution might be to explicitly model 3D symmetries but that is beyond the scope of 2D slot attention.

- Our experiments are entirely on synthetic data. This is a limitation of this paper and not of the model itself and an exciting direction for future work.

## Appendix B. Related Work

Prior object discovery approaches use an encoder-decoder framework with few exceptions (Greff et al., 2019; Kipf et al., 2020; Huang et al., 2020). The encoder processes images and populates latent slots, and the decoder uses the information about object present in latent slots to reconstruct the input. Some works employ spatial symmetries in the decoder (Eslami et al., 2016; Kosiorek et al., 2018; Crawford and Pineau, 2019; Lin et al., 2020; Jiang and Ahn, 2020; Monnier et al., 2021; Smirnov et al., 2021; Sauvalle and de La Fortelle, 2022) using the Spatial Transformer (Jaderberg et al., 2015), which is equivariant to $2D-$affine transformations. In contrast, most prior works use monolithic encoders (Greff et al., 2016; Eslami et al., 2016; Greff et al., 2017; Kosiorek et al., 2018; Burgess et al., 2019; Engelcke et al., 2020; Jiang and Ahn, 2020; Engelcke et al., 2021; Monnier et al., 2021; Smirnov et al., 2021; Emami et al., 2021) with only limited equivariance to translation in the case of convolutional encoders with output anchors (Crawford and Pineau, 2019; Lin et al., 2020). Alternatively, models iteratively *process* the encoded inputs to refine object detections (Locatello et al., 2020; Huang et al., 2020). Slot Attention (Locatello et al., 2020) appends absolute coordinates to feature maps in the encoder, thus making object detections sensitive to locations.

## Appendix C. Theoretical analysis of translation equivariance

We show that Slot Attention is equivariant to joint translation of the input features and the initial slot positions. We do not show equivariance to translations of individual objects due to occlusions, but a similar line of reasoning would otherwise work.

We formalize Slot Attention as a function with four inputs and two outputs:

$$\text{SA}(\text{inputs}, \text{abs\_grid}, S, S_p) = (S', S_p') \tag{3}$$

Here, inputs : $\mathbb{Z}^2 \to \mathbb{R}^{D_{\text{inputs}}}$ map feature coordinates to input vectors, abs\_grid : $\mathbb{Z}^2 \to \mathbb{R}^2$ is a linear function that maps feature coordinates to real-valued coordinate encodings,

$S \in \mathbb{R}^{K \times D_{\text{slots}}}$ are the initial latent slots and $S_p \in \mathbb{R}^{K \times 2}$ are the initial slot positions. Correspondingly, $S'$ and $S'_p$ are the final latent slots and slot positions after $T$ rounds of slot attention.

Next, we show that the final slot positions are equivariant to a joint translation of the input and initial slot positions, and that the final latent slots are invariant to said transformation:

$$\text{SA}(\text{inputs} \circ L_t, \text{abs\_grid}, S, R_t^{-1}(S_p)) = (S', R_t^{-1}(S'_p)) \tag{4}$$

Here, $t$ belongs to the group of translations over $\mathbb{Z}^2$, $L_t(x) = x + t$, $x \in \mathbb{Z}^2$, is a group action that translates an integer-valued coordinate and $R_t(y) = y + \text{abs\_grid}(t)$, $y \in \mathbb{R}^2$, is a translation in the space of real-valued coordinates. We use the inverse of $R_t$ because slot positions are used to re-center feature coordinate grids by inverting the translations applied to the input features.

We formalize the keys and values for the $k$th slot as mapping an integer-valued feature coordinate $x$ to a vector in $\mathbb{R}^D$.

$$\text{keys}^k(\text{inputs}, \text{abs\_grid}, S_p^k)(x) = f(k(\text{inputs}(x)) + g(\text{abs\_grid}(x) - S_p^k)) \tag{5}$$

$$\text{values}^k(\text{inputs}, \text{abs\_grid}, S_p^k)(x) = f(v(\text{inputs}(x)) + g(\text{abs\_grid}(x) - S_p^k)) \tag{6}$$

Next, we show that a joint translation of the inputs and the slot positions is equivalent to the translation of the key coordinates. The same can be shown for values.

$$\text{keys}^k(\text{inputs} \circ L_t, \text{abs\_grid}, R_t^{-1}(S_p^k))(x) \tag{7}$$

$$= f(k([\text{inputs} \circ L_t](x)) + g(\text{abs\_grid}(x) - R_t^{-1}(S_p^k))) \tag{8}$$

$$= f(k(\text{inputs}(x + t)) + g(\text{abs\_grid}(x) - (S_p^k - \text{abs\_grid(t)}))) \tag{9}$$

$$= f(k(\text{inputs}(x + t)) + g(\text{abs\_grid}(x) - S_p^k + \text{abs\_grid(t)})) \tag{10}$$

$$= f(k(\text{inputs}(x + t)) + g(\text{abs\_grid}(x + t) - S_p^k)) \tag{11}$$

$$= \text{keys}^k(\text{inputs}, \text{abs\_grid}, S_p^k)(x + t) \tag{12}$$

$$= [\text{keys}^k(\text{inputs}, \text{abs\_grid}, S_p^k) \circ L_t](x) \tag{13}$$

We use the assumption that abs\_grid is linear. Functions $f, g, k, v$ are applied per-position and we do not require linearity.

The cross attention between keys and slots (Algorithm 1, line 12) is computed for each pixel coordinate separately. Hence, it is trivially translation equivariant given the result for keys obtained above. The slot updates (line 13) are an attention-weighted sum over values. The sum is invariant to joint translations of both the attention mask and the values. The re-normalization of the attention mask for each slot on line 14 is also invariant to translations of the attention mask. We assume the sum to be finite.

Formally, we have the following properties hold for the attention mask computed on lines 12 and 14, and the updates computed on line 13:

$$\text{attn}^k(\text{inputs} \circ L_t, \text{abs\_grid}, S, R_t^{-1}(S_p^k)) = \text{attn}^k(\text{inputs}, \text{abs\_grid}, S, S_p^k) \circ L_t \tag{14}$$

$$\text{updates}^k(\text{attn}^k \circ L_t, \text{values}^k \circ L_t) = \text{updates}^k(\text{attn}^k, \text{values}^k) \tag{15}$$

Next, we compute the updated slot positions based on the attention mask (line 17):

$$S_p^{'k}(\text{attn}^k) = \sum_{x \in \mathbb{Z}^2} \text{attn}^k(x) * \text{abs\_grid}(x) \tag{16}$$

We show that translation equivariance holds for the updated positions.

$$S_p^{'k}(\text{attn}^k \circ L_t) = \sum_{x \in \mathbb{Z}^2} [\text{attn}^k \circ L_t](x) * \text{abs\_grid}(x) \tag{17}$$

$$= \sum_{x \in \mathbb{Z}^2} \text{attn}^k(x + t) * \text{abs\_grid}(x) \tag{18}$$

$$= \sum_{x \in \mathbb{Z}^2} \text{attn}^k(x) * \text{abs\_grid}(x - t) \tag{19}$$

$$= \sum_{x \in \mathbb{Z}^2} \text{attn}^k(x) * (\text{abs\_grid}(x) - \text{abs\_grid}(t)) \tag{20}$$

$$= [\sum_{x \in \mathbb{Z}^2} \text{attn}^k(x) * \text{abs\_grid}(x)] - \text{abs\_grid}(t) \tag{21}$$

$$= R_t^{-1}(S_p^{'k}(\text{attn}^k)) \tag{22}$$

Finally, since updates are invariant to the input transformation, lines 21 and 22 are also invariant. Hence, the updated latent slots $S'$ are invariant to the input transformation. The equivariant of $S_p^k$ and invariance of $S'$ holds over multiple iterations of Algorithm 1.

## Appendix D. Pseudo-Code

Algorithm 1 gives the self-explanatory pseudo-code of our method. In line 7 we scale the relative grid using $\delta$ after adjusting it using $S_s$ as otherwise for small objects, rel_grid will have numerically large values which we found made model training difficult.

## Appendix E. Additional results

### E.1. Out-of-distribution generalization on ObjectsRoom

Slot Attention uses the same mechanism to segment the foreground and the background. We test the interaction between translation and scale equivariance and multi-segment backgrounds in ObjectsRoom (Eslami et al., 2018), Figure 3. We find that both T-SA and TS-SA are more likely to learn the correct segments of the background (two walls, ground and ceiling), leading to between 10 and 15% absolute improvement in ARI.

We find that equivariant Slot Attention is robust to all out-of-distribution test sets, whereas the baseline deteriorates in the Empty Room OOD set due to increased over-segmentation of the background, see Table 2.

### E.2. Robustness to data augmentation on CLEVR

Given enough parameters and a training dataset that covers all spatial configurations a powerful deep learning model could potentially learn to be equivariant to spatial transformations at the object level. Data augmentation is typically used to augment the training set

---

**Algorithm 1:** Translation and Scale Equivariant Slot Attention

---

**Input:** inputs $\in \mathbb{R}^{N \times D_{inputs}}$, abs_grid $\in \mathbb{R}^{N \times 2}$, slots $\in \mathbb{R}^{K \times D_{slots}}$, Slot positions, $S_p \in \mathbb{R}^{K \times 2}$, Slot scales, $S_s \in \mathbb{R}^{K \times 2}$, $T$ iterations, small $\epsilon$.

**Data:** Encoders $f, g, k, v, q$, parameters of LayerNorms, MLP and GRU, $\delta$

**Output:** slots $\in \mathbb{R}^{K \times D_{slot}}$, $S_p \in \mathbb{R}^{K \times 2}$, $S_s \in \mathbb{R}^{K \times 2}$.

**1** inputs = LayerNorm(inputs)
**2 for** $t = 1$ **to** $T + 1$ **do**
**3**     slots_prev = slots
**4**     slots = LayerNorm(slots)
**5**
**6**     # Computes relative grids per slot, and associated key, value embeddings.
**7**     rel_grid = (abs_grid $- S_p$) / $S_s \times \delta$
**8**     keys = $f\left(k(\text{inputs}) + g(\text{rel\_grid})\right)$
**9**     values = $f\left(v(\text{inputs}) + g(\text{rel\_grid})\right)$
**10**
**11**     # Inverted dot production attention.
**12**     attn = softmax($\frac{1}{\sqrt{K}}$keys $* q(\text{slots})^T$, axis = "slots")
**13**     updates = WeightedMean(weights = attn, values = values)
**14**
**15**     # Updates $S_p$, $S_s$ and slots.
**16**     $S_p$ = WeightedMean(weights = attn, values = abs_grid)
**17**     $S_s = \sqrt{\text{WeightedMean(weights = attn} + \epsilon, \text{values = (abs\_grid} - S_p)^2)}$
**18**     **if** $t < T + 1$ **then**
**19**        slots = GRU(state = slots_prev, inputs = updates)
**20**        slots += MLP(LayerNorm(slots))
**21**     **end**
**22 end**

---

| Method | ARI (11 slots) | | | |
| --- | --- | --- | --- | --- |
| | **Validation** | **Six Objects** | **Empty Room** | **Identical Colors** |
| SA | $71.2_{\pm 16.6}$ | $71.3_{\pm 16.8}$ | $65.9_{\pm 14.8}$ | $69.6_{\pm 16.6}$ |
| T-SA | $78.7_{\pm 3.8}$ | $79.7_{\pm 2.4}$ | $71.1_{\pm 6.5}$ | $78.4_{\pm 2.7}$ |
| TS-SA | $87.1_{\pm 7.5}$ | $85.0_{\pm 5.0}$ | $86.4_{\pm 9.9}$ | $86.1_{\pm 6.3}$ |

Table 2: ARI in ObjectsRoom validation set and three out-of-distribution test sets, 11 slots.

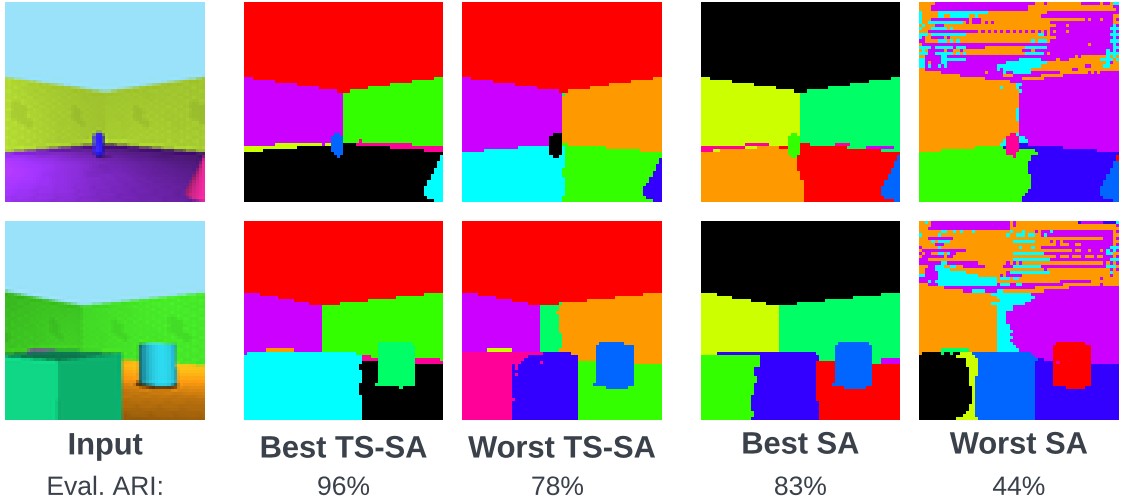

|  | Input | Best TS-SA | Worst TS-SA | Best SA | Worst SA |
|---|---|---|---|---|---|
| Eval. ARI: |  | 96% | 78% | 83% | 44% |

Figure 3: Segmentation mask examples for the best and worst translation and scale equivariant Slot Attention (TS-SA) and baseline Slot Attention (SA) out of five random seeds. TS-SA is less likely to over-segment the backgrounds and avoids the failure mode shown in the right-most column. We also report ARI over the entire validation dataset for each model.

| Method | FG-ARI | | |
|---|---|---|---|
|  | CLEVR | CLEVR Augm. Eval. | CLEVR Augm. Train. & Eval. |
| SA | $99.0_{\pm 0.2}$ | $93.6_{\pm 2.2}$ | $97.3_{\pm 0.4}$ |
| TS-SA | $98.9_{\pm 0.1}$ (-0.1) | $95.9_{\pm 1.0}$ (+2.3) | $98.4_{\pm 0.8}$ (+1.1) |
| SPACE | $40.1_{\pm 20.3}$ | $39.1_{\pm 19.7}$ | $44.4_{\pm 24.0}$ |

Table 3: FG-ARI in CLEVR with cropping and scaling data augmentation applied either only to the validation set or to both the training and the validation set.

with all possible spatial variations at the image level hoping to achieve some of this effect. We analyze, on the CLEVR-10 dataset, the effect of data augmentation during training and evaluation.

Baseline SA achieves close to 99% FG-ARI on this dataset. However, as seen in Table 3, under the column "CLEVR Augm. Eval.", this model is not robust to perturbations in the translation and scale of input images at test time. We hypothesize that SA tends to overfit to the objects appearing in the central portion of the images as well as to the constrant background. In column 4, "CLEVR Augm. Train. & Eval.", we find that model performance is not restored simply by using data augmentation during training. Details of the augmentation used are discussed in Appendix F.

On the other hand, our TS-SA model is relatively robust to test time augmentation and additionally benefits from global image level data augmentation during training suggesting that these two changes are somewhat orthogonal. We conclude that the inductive biases facilitated by equivariance cannot be supplanted by data augmentation alone.

## Appendix F. Datasets and data preprocessing

We use the standard pre-processing pipeline in CLEVR (e.g. Locatello et al. (2020)). In the data augmentation experiment, we sample random square crops that cover at least 25% of the original unprocessed image; these crops are then resized to $64\times64$, as in the original data processing. For CLEVRTex, we use the data processing from Karazija et al. (2021). We do not perform any data preprocessing for Tetrominoes and objects_room. RGB values in all datasets are scaled to $[0, 1]$. In the Tetrominoes dataset, in Figure 2 (right), we sample random square crops with a minimum area coverage tuned to be optimal for the baseline. The same setting is them used for our method.

## Appendix G. Model architectures and hyper-parameters

We use the same encoder (Table 6) and decoder (Table 7) on objects_room, CLEVR and CLEVRTex. In CLEVRTex, we also use a ResNet-34 encoder. The ResNet is not pre-trained and the downsampling before its first stage is removed (the first convolutional layer's stride is set to 1 and the max-pooling layer is removed). In Tetrominoes, we do not downsample in the encoder (Table 5) and we use a per-pixel decoder (Table 8), similarly to Kabra et al. (2021).

Different from the original Slot Attention, we use learnable initial latent slots (Kipf et al., 2022; Elsayed et al., 2022), which usually lead to better results. Initial slot positions are randomly sampled from $U(-1, 1)$ in both the x and y axes, and initial slot scales are sample from $N(0.1, 0.1)$ and clipped between 0.01 and 5. We did **not** perform hyper-parameters search for the initialization.

Hyper-parameters are reported in Table 4. All Slot Attention models are trained for 500k steps without early stopping or model selection. In the Tetrominoes experiment with limited dataset sizes, we found it sufficient to train for only 20k steps.

| Name | Value |
|---|---|
| attention $\epsilon$ | $10^{-8}$ |
| T | 3 |
| Adam: learning rate | $4 * 10^{-4}$ |
| Adam: $\beta_1$ | 0.9 |
| Adam: $\beta_2$ | 0.999 |
| Adam: $\epsilon$ | $10^{-8}$ |
| Warm-up steps | 10k |
| Learning rate schedule | Cosine decay |
| Slot dim. | 64 |
| $k, v, q, g$ | Linear(128) |
| $f$ | MLP(128, 1 hidden layer, ReLU) |

Table 4: Small/standard Slot Attention.

| Type | Size/Channels | Activation | Comment |
|---|---|---|---|
| Conv 5×5 | 64 | ReLU | stride: 1 |
| Conv 5×5 | 64 | ReLU | stride: 1 |
| Conv 5×5 | 64 | ReLU | stride: 1 |
| Conv 5×5 | 64 | ReLU | stride: 1 |

Table 5: CNN encoder, used in Tetrominoes experiments.

| Type | Size/Channels | Activation | Comment |
|---|---|---|---|
| Conv 5×5 | 64 | ReLU | stride: 2 |
| Conv 5×5 | 64 | ReLU | stride: 2 |
| Conv 5×5 | 64 | ReLU | stride: 1 |
| Conv 5×5 | 64 | ReLU | stride: 1 |

Table 6: CNN encoder.

| Type | Size/Channels | Activation | Comment |
|---|---|---|---|
| Spatial Broadcast | 16×16 | - | - |
| (Relative) Position Encoding | Slot Dim. | - | - |
| Transposed Conv 5×5 | 64 | ReLU | stride: 2 |
| Transposed Conv 5×5 | 64 | ReLU | stride: 2 |
| Conv 5×5 | 64 | ReLU | stride: 1 |
| Conv 5×5 | 64 | ReLU | stride: 1 |
| Conv 1×1 | 4 | - | stride: 1 |
| Split Channels | RGB(3), alpha mask(1) | Softmax (on alpha mask) | - |
| Recombine Slots | - | - | - |

Table 7: Spatial Broadcast Decoder with a CNN.

| Type | Size/Channels | Activation | Comment |
|---|---|---|---|
| Spatial Broadcast | 35×35 | - | - |
| (Relative) Position Encoding | - | - | - |
| Conv 1×1 | 256 | ReLU | stride: 1 |
| Conv 1×1 | 256 | ReLU | stride: 1 |
| Conv 1×1 | 256 | ReLU | stride: 1 |
| Conv 1×1 | 256 | ReLU | stride: 1 |
| Conv 1×1 | 256 | ReLU | stride: 1 |
| Conv 1×1 | 4 | - | stride: 1 |
| Split Channels | RGB(3), alpha mask(1) | Softmax (on alpha mask) | - |
| Recombine Slots | - | - | - |

Table 8: Spatial Broadcast Decoder with an MLP.

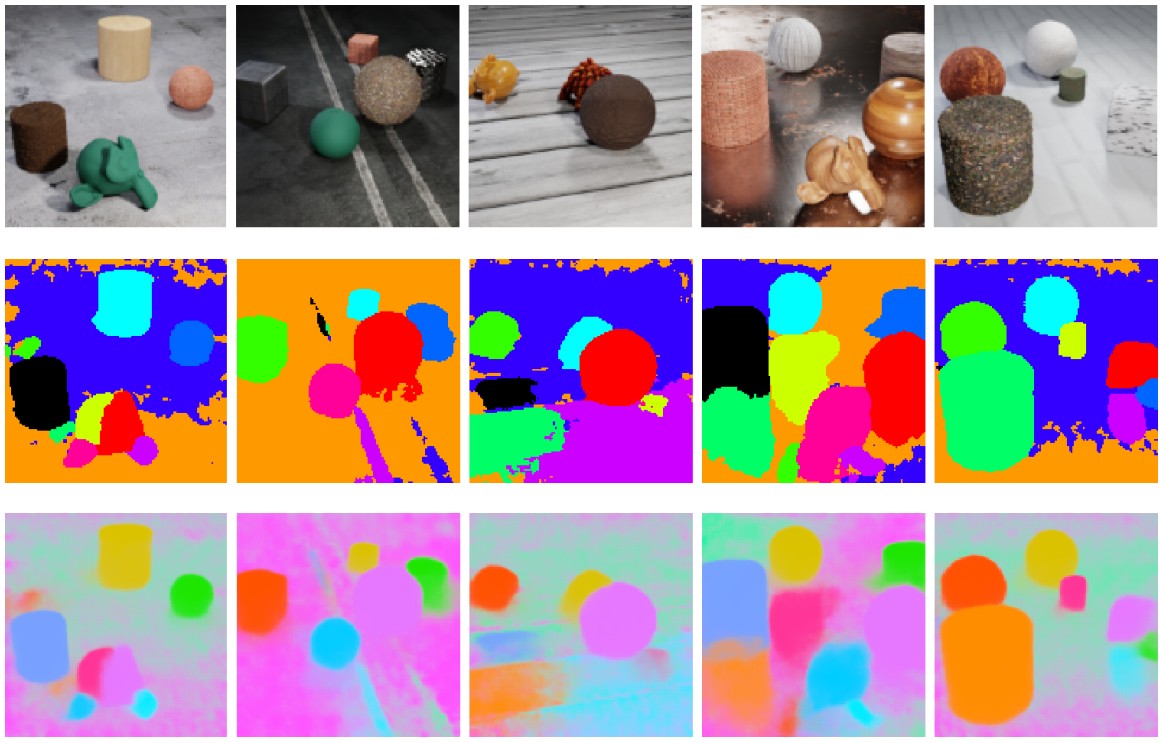

Figure 4: Examples of segmentation masks for ResNet TS-SA trained on CLEVRTex. Top: original images, middle: predicted segmentation mask with per-pixel argmax, bottom: predicted segmentation mask with uncertainty.

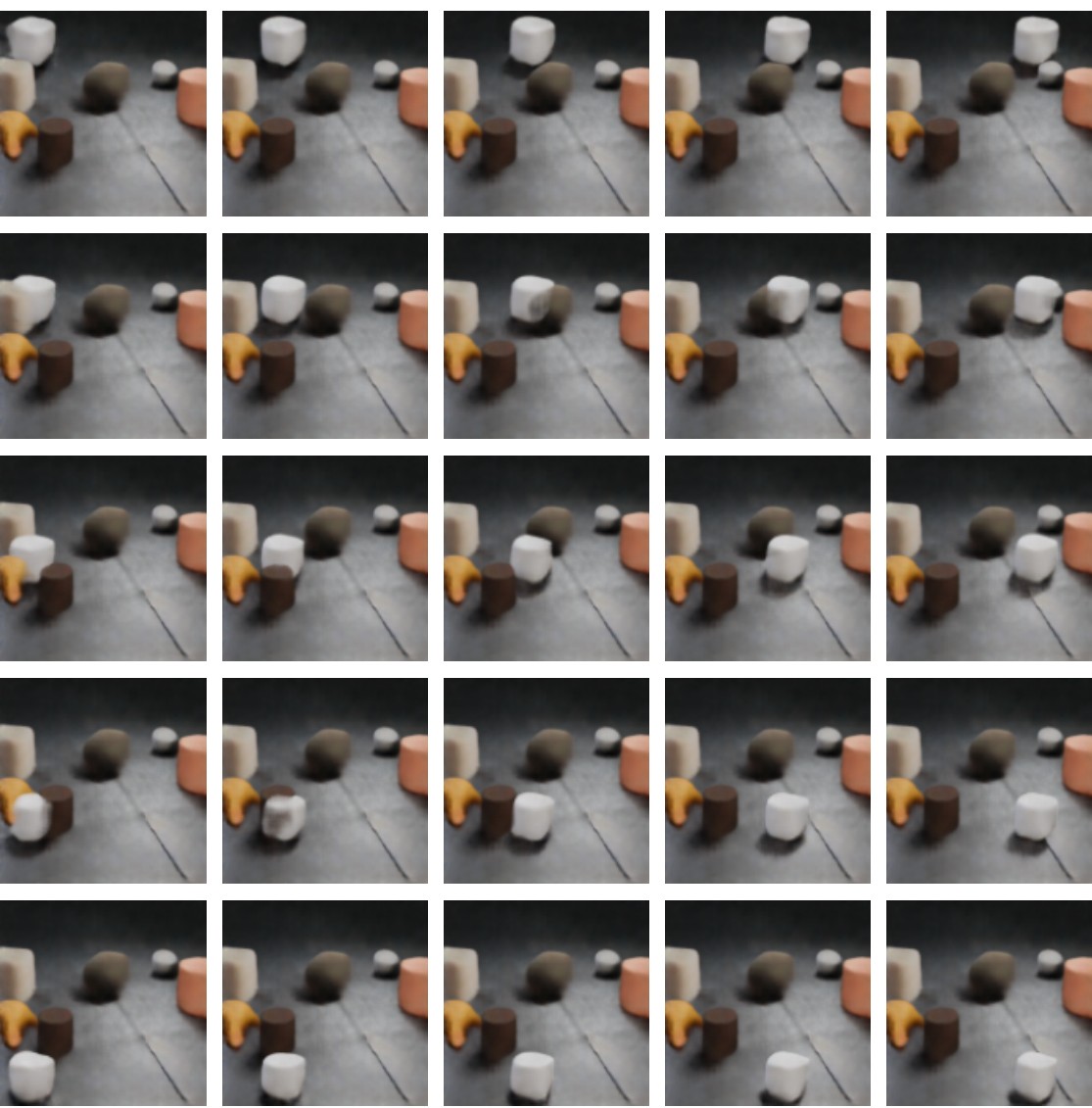

Figure 5: Changing position of a slot representing the white cube.

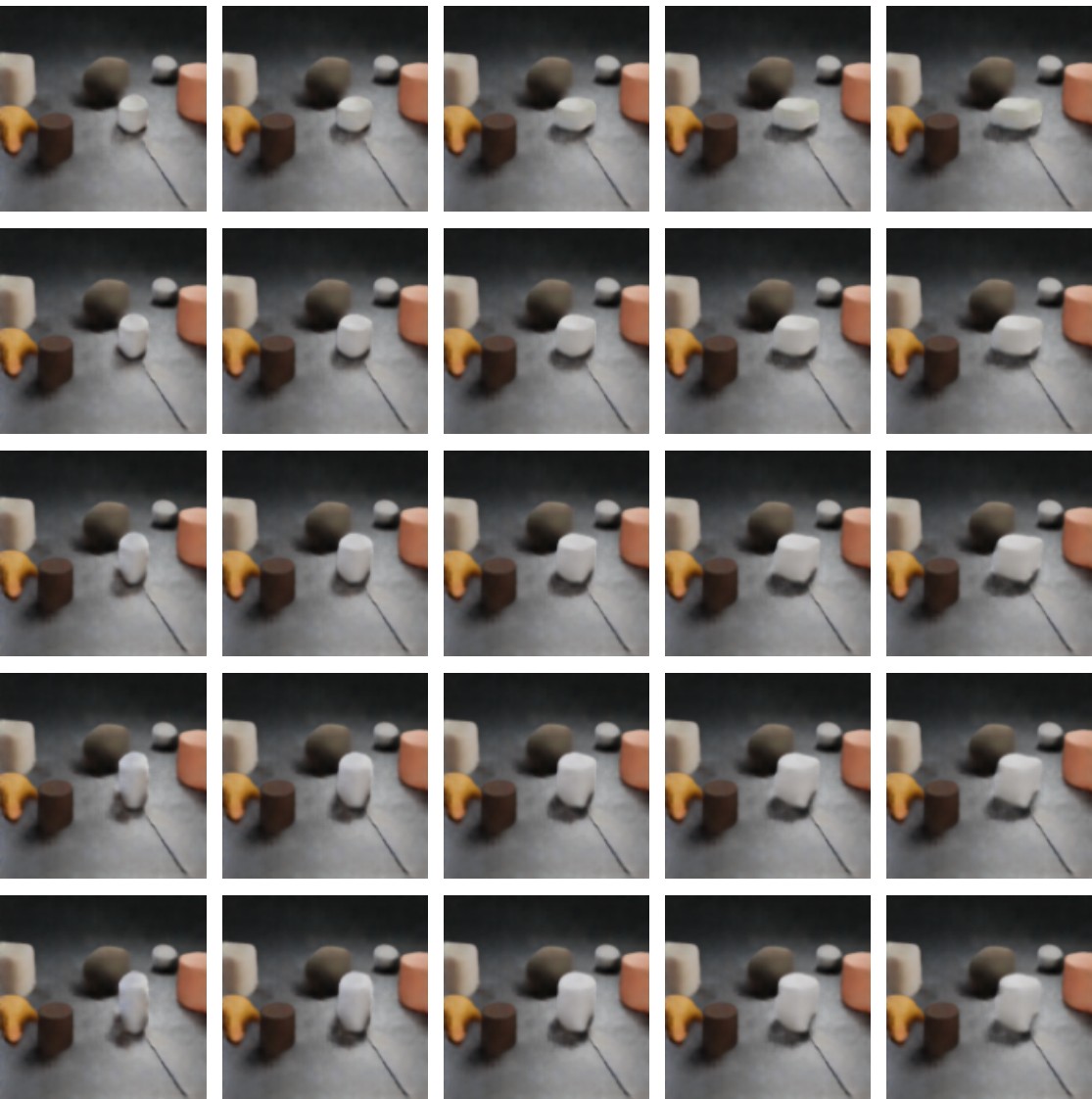

Figure 6: Changing scale of a slot representing the white cube.

