# OpenReview forum: "Spatial Symmetry in Slot Attention"
_NeurIPS.cc/2022/Workshop/NeurReps — NeurReps 2022 Poster_

### Official Review · Reviewer_pKAH · 2022-10-11
**Weak Relevance; Better fit in a CV venue.**

**Confidence:** 4
**Soundness:** 2
**Presentation:** 2
**Contribution:** 2
**Overall Rating:** 4

**Summary:**

The authors propose an explicit encoding of position and scale in attentional slot-based neural networks.
They demonstrate how this addition can result in large gains in data efficiency and scene decomposition performance, in a self-supervised object centric setup.

**Questions:**

Why not title the work: "Spatial Invariance in Slot Attention"?



**Limitations:**

The authors did not discuss the limitations of their explicit spatial encodings.

**Recommended Decision:**

1: Reject

**Relevance:**

2: Limited relevance

**Strengths And Weaknesses:**

++ Informative overview of related work.

++ Promising experimental results

-- Low relevance to the workshop: There is no focus on the geometry and topology of the representations learned or of the model's behavior. The notion of symmetry presented is actually about invariance to position and scale, not about symmetry in the representations learned. Overall, the current manuscript is a better fit for a Computer Vision venue, than to NeurReps.

-- The Computer-Vision task addressed is not properly described.

**Submission Track:**

Extended Abstract (4 Page)

---

### Official Review · Reviewer_9peU · 2022-10-17
**Adding and testing equivariance(s) in slot attention**

**Confidence:** 3
**Soundness:** 3
**Presentation:** 4
**Contribution:** 3
**Overall Rating:** 6

**Summary:**

The authors propose a method for building in spatial symmetries into Slot Attention modules of neural networks. More specifically, they incorporate translation and/or scale equivariance into Slot Attention and the Spatial Broadcast decoder. The authors then test their method on 4 datasets and demonstrate improvements in terms of performance and training efficiency.

**Questions:**

Are there examples (that you have directly tested, or can imagine) where SA would perform miserably and T-SA or TS-SA would be required to accomplish the task? I'd like to understand better whether the contribution is primarily one of efficient training/faster generalization, or also one that can solve more difficult classes of problems.

I'm not sure if it's correct that in Locatello et al. (2020), the authors use an N×2 "abs_grid". In that paper, the authors state, "each point on the grid is associated with a **4-dimensional feature vector** that encodes its distance (normalized to [0, 1]) to the borders of the feature map along each of the four cardinal directions" (emphasis added). Can you please clarify this point for me?

Can you please provide further details as to how positions and scales were manipulated (in Figures 1, 5, 6) to generate the different images?



**Limitations:**

The list of limitations is reasonable and upfront. Questions about other possible limitations have been implicitly addressed in other parts of this review.

**Recommended Decision:**

3: Accept

**Relevance:**

4: Highly relevant

**Strengths And Weaknesses:**

Overall my impression is that the impact of this paper (as the current results stand) is modest but that the paper aligns rather well with the workshop's goal of exploring equivariant representations and has enough contribution to be appropriate for the Extended Abstract track.

Strengths:

- Although the modification to slot attention is relatively small, the authors are upfront about this and make reasonable efforts to describe their contribution in light of the surrounding literature.
- The paper is easy to read and provides good motivations and contextualization for results.
- The appendices also provide helpful supporting information (pseudocode, limitations, additional qualitative examples).

Weaknesses:

- It's unclear whether T-SA or TS-SA provide significant improvements over SA at scale (large datasets with longer training time). To my understanding, the improvements shown in Figure 1 above baseline are relatively small.
- The captions for the figures in the main paper could be more descriptive (they seem to describe the context surrounding the figures, rather than the figures themselves). Given the main text of the paper, it works out okay, but the exposition would be clearer if the figures and their captions could stand better alone.

**Submission Track:**

Extended Abstract (4 Page)

---

### Official Review · Reviewer_ZoVp · 2022-10-18
**Solid paper with convincing results, well written**

**Confidence:** 3
**Soundness:** 4
**Presentation:** 4
**Contribution:** 3
**Overall Rating:** 8

**Summary:**

The paper introduce equivariance to translation and scale into the attention mechanism of Slot Attention proposed by  Locatello et al. 2020 and in the generation mechanism of Spatial Broadcast Decoder proposed by Watters et al. 2019. Experiments show that adding inductive biases to Slot Attention improves significantly its performance in 4 datasets.

**Questions:**

Why does Table 1 do not include the results you mentioned from Sauvalle et al. 2022? Their FG-ARI improves significantly over your best results by ~10%. If the reason is because they used a pretrained model, what prevented you to use pretrained models as well?

**Limitations:**

Appendix A covers the limitations with a thorough analysis.

**Recommended Decision:**

3: Accept

**Relevance:**

3: Solid fit

**Strengths And Weaknesses:**

Strenghts: The paper is solid. It has convincing results improving significantly over the baseline (Slot Attention). Experiments are well done, cover 4 different datasets with complementary challenges, and contain both ablation analysis and qualitative results. The appendix is properly done and covers various aspects, from proofs to pseudocode, facilitating reproducibility.

**Submission Track:**

Extended Abstract (4 Page)

---

### Decision · Program_Chairs · 2022-10-21

Accept (Poster)